# Hyperprogression under Immunotherapy

**DOI:** 10.3390/ijms20112674

**Published:** 2019-05-30

**Authors:** Maxime Frelaut, Christophe Le Tourneau, Edith Borcoman

**Affiliations:** 1Department of Drug Development and Innovation (D3i), Institut Curie, Saint-Cloud, 75005 Paris, France; maxime.frelaut@curie.fr (M.F.); christophe.letourneau@curie.fr (C.L.T.); 2INSERM U900 Research unit, Institut Curie, 92210 Saint-Cloud, France; 3Paris-Saclay University, 75005 Paris, France

**Keywords:** immunotherapy, new patterns of response, pseudoprogression, treatment beyond progression, hyperprogression

## Abstract

Immunotherapy is now widely prescribed in oncology, leading to the observation of new types of responses, including rapid disease progression sometimes reported as hyperprogression. However, only a few studies have assessed the question of hyperprogression and there is no consensual definition of this phenomenon. We reviewed existing data on hyperprogression in published studies, focusing on reported definitions, predictive factors, and potential biological mechanisms. Seven studies retrospectively assessed hyperprogression incidence, using various definitions, some based on the tumoral burden variation across time with repeated computed-tomography (CT) scan, others based on an association of radiological and clinical criteria. Reported hyperprogression incidence varied between 4% and 29% of all responses, mostly in multi-tumor cohorts and with patients receiving immune checkpoint inhibitors. Hyperprogression correlated with worse chances of survival than standard progression in two studies. However, no strong predictive factors of hyperprogression were identified, and none were consistent across studies. In total, hyperprogression is a frequent pattern of response under immunotherapy, with a strong impact on patient outcome. There is a need for a consensual definition of hyperprogression. Immunotherapy should be stopped early in cases where there is suspicion of hyperprogression.

## 1. Introduction

Immunotherapy has represented for a few years a breakthrough in oncology, with the development of immune checkpoint inhibitors (ICI) targeting cytotoxic T-lymphocyte-associated antigen 4 (CTLA-4) and programmed cell death protein 1 (PD-1) or its ligands programmed cell death ligand 1 (PD-L1) and 2 (PD-L2) [1]. An increase in overall survival (OS) with these treatments has been demonstrated in a broad range of advanced cancer types including melanoma [2,3], non-small-cell lung cancer (NSCLC) [4,5,6], renal cancer [7], and head and neck squamous cell carcinoma (HNSCC) [8]. These treatments are now widely used in clinical practice.

By restoring an efficient antitumor T-cell response, ICI have been associated with a new pattern of responses, which have not been previously described with chemotherapy or targeted therapy [9], such as durable responses that may persist even after ICI interruption and pseudoprogression, described as an objective response following initial disease progression, which would be considered to be a disease progression using the current version of response-evaluation criteria in solid tumors (RECIST) [10]. Based on these observations, new specific response criteria have been developed, including immune-related response criteria (irRC) [11], immune-related response-evaluation criteria in solid tumors (irRECIST) [12], and immune RECIST (iRECIST) [13], defining the concept of unconfirmed progressive disease (uPD) that may be confirmed by a new radiological evaluation up to 12 weeks later.

Other studies have also reported unexpected rapid disease progressions under immunotherapy, called hyperprogressions, suggesting that these treatments could have a deleterious effect and may lead to cancer cell proliferation or acceleration of progression pace [14,15]. These hyperprogressions have been less studied and there is no consensual definition of this phenomenon. In addition, the mechanism of hyperprogression and potential predictive factors are still largely unknown. Even if the incidence of this phenomenon is uncertain, its comprehension is fundamental to precociously identifying patients experiencing hyperprogression and rapidly switching them onto another potential effective treatment.

Here, we will review existing data on hyperprogression, focusing on its definitions, reported predictive factors and potential mechanisms, and discuss clinical implications and practical considerations for the management of hyperprogressive patients.

## 2. Definitions and Reported Incidence of Hyperprogression

Hyperprogression was first described in case reports and retrospective studies of patients treated with ICI, with the observation that some cancer patients seem to have accelerated tumor growth after initiation of immunotherapy [14,16]. These data were consistent with OS curves from randomized trials, in which OS were better with chemotherapy than immunotherapy in the first weeks/months of treatment and then curves crossed, suggesting that ICI did worse than chemotherapy in a subgroup of patients [4,17]. See Figure 1 and Figure 2 illustrating the phenomenon of hyperprogression.

Several assessments of hyperprogression have been reported in retrospective studies to evaluate its incidence, and are summarized in Table 1. In a retrospective multi-tumor cohort of 131 patients treated in phase 1 studies with anti-PD-1 or anti-PD-L1, Champiat et al. extrapolated the tumor volume, assuming it could be approximated by a sphere with a diameter equal to the sum of larger diameters of target lesions (according to RECIST), and studied its evolution over time, with two pretherapeutic computed-tomography scans (CT scan), one 8 weeks before baseline CT scan and one at baseline, and one after initiation of immunotherapy, to estimate the tumor growth rate (TGR) reported as the percentage increase in tumor volume per month [14]. An 8-week washout period of antitumoral treatment was required before initiation of immunotherapy to use each patient as his/her own control. The authors defined hyperprogressive patients as having a RECIST progression at the first evaluation and at least two-fold TGR increase between the pretherapeutic and the immunotherapy period, and observed a rate of 9% of hyperprogressions (Figure 3).

In the same way, Ferrara et al. compared pre- and post-immunotherapy TGR in a retrospective cohort of 406 patients with advanced NSCLC treated with PD-1/PD-L1 inhibitors, defining hyperprogressive disease as a RECIST progression at the first evaluation and a difference between on-treatment and pre-treatment TGR (∆TGR) exceeding 50% [18]. They reported that 13.8% of patients experienced hyperprogression.

Kanjanapan et al. used the same method to compute TGR, defining hyperprogression as RECIST progression at the first evaluation and more than two-fold increase of TGR after initiation of immunotherapy [19]. Among 182 patients treated with immunotherapy in multi-tumor phase 1 studies (including single-agent and immunotherapy combinations), they retrospectively observed 7% of hyperprogressive disease.

In a retrospective cohort of 34 patients with advanced HNSCC treated with PD-1/PD-L1 inhibitors, Saâda-Bouzid et al. used the tumor growth kinetics (TGK), which is the difference of the sum of the largest diameters of the target lesions (according to RECIST) per unit of time between two evaluations [15]. Hyperprogression was defined as TGK ratio (ratio of TGK after immunotherapy to pretherapeutic TGK) ≥2, with an observed hyperprogression rate of 29%.

More recently, Russo et al. reported the results of their retrospective analysis of 152 NSCLC patients treated with ICI, defining hyperprogression as RECIST progression associated with at least three of these criteria: time to treatment failure (TTF, time from the start of treatment to discontinuation for any reason) less than two months; ≥50% increase of the sum of target lesion major diameters between baseline and first radiological evaluation; appearance of at least two new lesions in an organ already involved between baseline and first radiological evaluation; spread of the disease to a new organ between baseline and first radiological evaluation; clinical deterioration with decrease in Eastern Cooperative Oncology Group (ECOG) performance status ≥2 during the first 2 months of treatment [20]. Thirty-nine patients (25.7%) met these criteria and were defined as experiencing disease hyperprogression.

Kato et al. retrospectively analyzed 155 patients with metastatic cancer treated by immunotherapy [21]. They defined hyperprogression as a TTF < 2 months, >50% increase in tumor burden compared to pre-immunotherapy imaging and more than two-fold increase in progression pace, using immune-related criteria. The authors found that 6 of the 155 patients (4%), including all types of tumors, experienced hyperprogression.

Finally, Matos et al. focused on phase 1 patients treated with ICI, defining hyperprogressive disease as TTF less than 2 months, minimum increase in measurable lesions of 10 mm and ≥40% increase in target tumor burden compared to baseline or ≥20% increase plus the appearance of multiple new lesions, based on RECIST [22]. Over 214 patients, they observed hyperprogression in 15% of cases.

Interestingly, patients experiencing hyperprogression under immunotherapy had a statistically worse OS than patients with standard progression in Ferrara’s (3.4 vs. 6.2 months, *p* = 0.003) [18] and Matos’s cohorts (4.8 vs. 8.7 months, *p* = 0.03) [22]. There was a trend toward the same association in Champiat’s (4.6 vs. 7.6 months, *p* = 0.19) [14] and Saâda-Bouzid’s cohorts (6.1 vs. 8.1 months, *p* = 0.77) [15].

Hyperprogression under immunotherapy has been reported by several distinct teams and could have a deleterious survival effect on patients, which needs to be considered. Hence, identifying predictive factors of hyperprogression would certainly help clinicians to evaluate the benefit–risk ratio before starting immunotherapy.

## 3. Predictive Factors

The above-mentioned studies reporting hyperprogression under immunotherapy enlightened some predictive factors of hyperprogression (Table 1).

Analyzing the 12 patients experiencing hyperprogression in their multi-tumor phase 1 cohort, Champiat et al. did not observe any association between hyperprogression and tumor burden, previous treatment line, and histology [14]. Hyperprogressive disease was also independent of type of ICI (PD-1 vs. PD-L1 antibodies) and tumoral PD-L1 status. However, patients with hyperprogressive disease were significantly older than other patients (66 vs. 55 years old, *p* = 0.007), and among the 36 patients older than 65 years, hyperprogression was more frequent than among younger patients (19% vs. 5%, *p* = 0.018).

Ferrara et al. did not observe any association between hyperprogression and age (with 46% of patients older than 65 years old), tumoral PD-L1 status, molecular status including *EGFR* alterations, previous treatments, type of ICI (anti-PD-1 vs. anti-PD-L1), and ECOG status among 406 patients with advanced NSCLC treated with PD-1/PD-L1 inhibitors [18]. The authors found that hyperprogression was more frequent among patients who had more than two metastatic sites (*p* = 0.006).

In a similar population and with almost the same definition of hyperprogressive disease than Champiat et al., Kanjanapan et al. did not observe any correlation between hyperprogression and age [19]. There was no association with performance status, tumor type, tumor volume, type of immunotherapy (combination vs. monotherapy), number of previous lines, or immune-related toxicity reported in their study. However, it was the only study where hyperprogression was associated with gender, with more hyperprogressive disease among women (*p* = 0.01).

Among patients with HNSCC, hyperprogressive diseases after immunotherapy initiation were more frequent in cases of locoregional recurrence (37% vs. 9%, *p* = 0.008) [15], and almost all cases of hyperprogression occurred in patients with a recurrence in the irradiated field. In this study, the authors did not observe any statistical association between hyperprogression and age, tumor volume, number of previous lines, PD-1 vs. PD-L1 inhibitors, tobacco exposure, or HPV tumoral status.

Kato et al. investigated genomic alterations that could be associated with immunotherapy outcome and hyperprogression, using next-generation sequencing (NGS) available data of 155 multi-tumor patients [21]. A favorable outcome after immunotherapy (defined as TTF ≥ 2months) was observed for patients with *TERT* (OR: 0.42; *p* = 0.07), *PTEN* (OR: 0.28; *p* = 0.10), *NF1* (OR: 0.15; *p* = 0.07) and *NOTCH1* (OR: <0.19; *p* = 0.02) gene alterations. In contrast, *EGFR* (OR: 10.2; *p* = 0.002), *MDM2/4* (OR: >11.9; *p* = 0.001) and *DNMT3A* (OR: 9.33; *p* = 0.03) alterations were associated with TTF < 2 months. In multivariate analysis, melanoma was significantly associated with a longer TTF (*p* = 0.02) compared to other cancer types, while *EGFR* (*p* = 0.02), *MDM2/4* (*p* = 0.02) and *DNMT3A* (*p* = 0.04) alterations were associated with a worse outcome. Four of the six patients with *MDM2/4* amplifications (67%) and two of the ten patients with *EGFR* alterations (20%) experienced hyperprogression. Among patients with *DNMT3A* alteration (five patients), only one was radiologically evaluable and did not experience hyperprogressive disease.

In their study, Russo et al. compared histopathological and molecular results of 35 advanced NSCLC patients (with 12 patients who experienced hyperprogression) with tissue samples available and evaluable responses after immunotherapy [20]. They did not observe significant differences between clinical outcomes according to density in tumor-infiltrating T lymphocytes (TILs) of CD4^+^/CD8^+^ lymphocytes, regulatory T-cells (Tregs) (based on FOXP3 expression), peritumoral and stromal myeloperoxidase myeloid cells, and PD-1^+^ and PD-L1^+^ immune cells. However, hyperprogression was correlated with the density of myeloperoxidase myeloid cells within the tumor (*p* = 0.05) and inversely correlated with PD-L1 expression in tumor cells (*p* = 0.046). In all patients experiencing hyperprogression, a specific subtype of CD163^+^ CD33^+^PD-L1^+^ macrophages with an epithelioid morphology was observed, and this was statistically more frequent than in non-hyperprogressive patients, defining a population of tumor-associated macrophages (TAM) that was enriched in patients with hyperprogression (*p* < 0.0001). Presence of *MDM2/4* amplification was not associated with hyperprogression in their study.

Another interesting approach would be to identify pharmacodynamic biomarkers that could help diagnose patients experiencing hyperprogression earlier and avoid deleterious survival impact. In this way, Weiss et al. evaluated chromosomal instability using NGS on plasma/serum-derived cell-free DNA (cfDNA) as a marker of immunotherapy outcome [23]. They prospectively assessed the evolution of genomic copy number instability (CNI) between each cycle of treatment among 56 patients with multiple tumor types treated with immunotherapy or combination treatments with PD-1 inhibitors. The authors could accurately predict progression based on chromosomal instability quantification in plasma cfDNA, with a risk of progression over 90% in patients without a substantial decrease in the CNI score. Interestingly, in five of the six patients who experienced hyperprogression, progression was early predicted using CNI score. However, this pharmacodynamics monitoring will need to be further validated in large prospective studies. Data regarding predictive factors of hyperprogression are discordant between studies and for now no validated predictive factor of hyperprogression has been identified that can be applied in clinical routine.

## 4. Biological Rationale

A better biological understanding of hyperprogression could help identify predictive factors. However, few data exist regarding hyperprogression biological mechanisms.

Several primary and adaptive mechanisms of resistance to ICI have been described. Response to ICI seems to be conditioned by the infiltration of tumors by activated T-cells, witnessing an ongoing active antitumor immune response. Tumors that lack TILs in the tumor bed, corresponding to so-called “cold tumors”, have been described as presenting low or no response to ICI [24,25,26]. The presence of immunosuppressive cells in tumor tissues such as Tregs [27], or other myeloid-derived suppressor cells [20] could also alter antitumor immunity.

Others mechanisms of resistance to ICI have been reported such as the absence of tumor recognition by T-cells, due to the lack of immunogenic tumor antigens (i.e., low mutational burden, absence of neoantigens or cancer-testis antigens), or to the development by the tumor of mechanisms that impair antigen presentation such as major histocompatibility complex down-regulation [28,29].

It has also been reported that alterations in tumor-intrinsic oncogenic pathways could impair the antitumor immune response, such as the activation of the PI3K/AKT/mTOR signaling pathway [30,31] or the WNT/β-catenin signaling pathway [32]. All these mechanisms of resistance should be further assessed in the context of hyperprogression.

Based on the clinical observations described above, Russo et al. sought to evaluate the role of macrophages implicated in detrimental effects under ICI treatment [20]. After implantation of NSCLC cells in nude mice, they treated mice with anti-mouse PD-1 antibody and observed an increased tumor growth compared with control, and enrichment in tumor micro-environment by macrophages similar to what they observed in hyperprogressive patients. Because blocking of PD-1 receptors in TAM has been described to restore antitumor functions [33], authors hypothesized that the detrimental effect of the antibody could be led by the crystallizable fragment (Fc) domain fixation to Fc receptor (FcR). Accordingly, they performed the same experiment with anti-PD-1 F(ab)_2_ fragments and observed that the lack of Fc domain abrogated the increased tumor growth effect of the full antibody. They further used two EGFR^+^ NSCLC patient-derived xenograft (PDX) models injected in immunodeficient mice, and treated mice with anti-human PD-1 (nivolumab). They observed a significant increase in tumor growth after exposition to nivolumab, associated with an accumulation of M2-like macrophages. Once again, the exposition to anti-human PD-1 F(ab)_2_ did not lead to a hyperprogressive-like growth in these models in comparison to the full antibody. With these results, the authors suggested that the interaction between Fc of ICI and FcR on specific M2-like intra-tumoral macrophages could lead to a reprogramming of these cells toward a pro-tumorigenic function, suggesting that innate immunity might be implicated in hyperprogression mechanisms.

Recently, Stein et al. investigated the relationship between TILs and tumoral cells [34]. Because infiltrating T-cells in breast cancer show a low expression of Granzyme B [35] and high expression of PD-1 [36], authors focused on cognate non-lytic antigen-specific interaction between CD8^+^ T lymphocytes and tumor cells. In vitro, they challenged breast cancer cells with specific CD8^+^ lymphocytes depleted of cytotoxic granula. Using microarrays, cognate non-lytic interactions induced expression of mediators of immune resistance (such as PD-L1, indoleamine-2,3-dioxygenase and Granzyme B inhibitor) and pluripotency-associated genes in cancer cells. This interaction also led to the enrichment of CD44^high^CD24^low^ tumor cells previously described as cancer stem cells (CSC) [37,38]. To assess the deleterious effect of these non-lytic cognate interactions, the authors inoculated mice with breast cancer cells after co-culture with specific or non-specific CD8^+^ lymphocytes depleted of cytotoxic granula. After 48 days, primary tumors were significantly larger after cognate than non-cognate co-culture (*p* = 0.03), and lymph node involvement was more frequent. Taken together, these results suggest that T-cells with loss of cytotoxic activity could reshape cancer cells after cognate interaction toward stemness cells. The authors hypothesized that in refractory tumors that have acquired the ability to modulate adaptive immunity, immunotherapy could promote these non-lytic interactions and lead through dedifferentiation of cancer cells to hyperprogression.

Recently, Kamada et al. assessed pre- and post-anti-PD-1 treatment tumor samples from patients with advanced gastric cancer who experienced hyperprogression [39]. They found that PD-1 blockade increased tumor-infiltrating proliferative effector Tregs, contrasting with their reduction in tumor samples of non-hyperprogressive patients. Under PD-1 blockade, they observed a significant enhancement of Tregs suppressive activity. In vivo, the genetic ablation of PD-1 or the antibody-mediated blockade of PD-1 in Tregs increased their proliferation and suppression of antitumor immune responses, suggesting that PD-1 blockade might facilitate the proliferation of highly suppressive PD-1+ effector Tregs in hyperprogressive patients, resulting in the inhibition of antitumor immunity.

These studies enlighten possible pathways leading to hyperprogression, suggesting that hyperprogression could be a real immunological phenomenon.

## 5. Controversies of the Hyperprogression Phenomenon

Hyperprogression has been described in several studies and should be now accepted as a real pattern of response under immunotherapy. This phenomenon is not rare, with patients experiencing hyperprogressive disease representing 4% up to 29% of the population in retrospective studies with different definitions [14,15,18,19,20,21,22]. According to Champiat and Ferrara, this rate could be underestimated, since some patients in their cohorts (respectively 8% and 30.5% of all patients) experienced rapid clinical deterioration with immunotherapy and could not been evaluated by CT scan and thus did not meet radiological criteria for hyperprogression [14,18]. Furthermore, there is no prospective data on hyperprogression rate for now. Hyperprogression have been observed in a wide range of tumor types; some of these studies were tumor type-specific for NSCLC [18,20] or HNSCC [15], but most included patients with various histology types [14,19,21,22]. All these studies reported no statistical variation of hyperprogression rate between cancer types. However, Kato et al. reported a longer TTF in melanoma patients compared to other types of cancer, which may suggest that hyperprogression is less frequent among melanoma patients, even if a short TTF is not enough to define hyperprogression and could be a sign of other events such as standard progression or limiting toxicity [21].

Even with increasing published data on hyperprogression in the literature, its definition is not consensual. Some authors used only radiological criteria, based on the variation of three-dimensional [14,18,19] or unidimensional [15] measurements of tumor burden over time to evaluate the rate of tumor growth (TGR and TGK) before and after immunotherapy initiation. These studies did not use the same cut-offs of tumor growth increase to define hyperprogression (2-fold [14,19] or ∆TGR > 50% [18]). This approach has some limitations. First, to compute TGR or TGK before immunotherapy, two CT-scan evaluations are needed, which could be limiting for some patients in the first line of treatment, for example. Furthermore, as described previously, CT scans could not be always performed in cases of hyperprogression because of rapid clinical deterioration, meaning that patients experiencing clinical hyperprogression cannot be considered to be real hyperprogressors based on these definitions. Third, TGR and TGK methods only evaluate the variation of target lesions and thus did not include new lesions in the assessment of tumor growth. Finally, these definitions could lead to a false classification of response pattern by only using radiological criteria: Ferrara et al. reported that almost 10% of hyperprogressive patients observed in their cohort were further reclassified as pseudoprogression [18]. On the other hand, some authors used a combination of clinical and radiological criteria [20,21,22]. Many of these studies defined hyperprogression disease with a TTF < 2 months (including progression, death, and treatment interruption for any reason) associated with radiological or clinical signs (Table 1). Even if these definitions included clinical criteria, they all mandated at least one radiological criterion and thus a post-immunotherapy CT scan, limiting their use in clinical practice and potentially underestimating hyperprogression frequency.

There are still investigations into whether hyperprogression is a real immunotherapy-specific phenomenon or just represents the natural evolution of cancer growth rate without any treatment. Rapid tumor flare has already been described in oncology, for example after tyrosine kinase inhibitor discontinuation [40,41,42]. To assess this question, Ferrara et al. used a chemotherapy control cohort to compare response patterns with immunotherapy and chemotherapy in NSCLC [18]. Fifty-nine patients with advanced NSCLC failing a platinum-based regimen and treated with mono-chemotherapy were included in a control cohort and were evaluated identically to patients in the immunotherapy cohort, with a comparison of TGR before and after new chemotherapy regimen initiation. Among the chemotherapy cohort, 3 patients (5%) experienced disease progression classified as hyperprogression vs. 13.8% in the immunotherapy cohort, meaning that this phenomenon is not specific to immunotherapy, although more frequent with immunotherapy. However, another interpretation could be that chemotherapy is more rapidly efficient than immunotherapy, and could slow down the natural growth rate acceleration in the first weeks of treatment, unlike immunotherapy. Hyperprogression could be fortuitously observed concomitantly to the onset of immunotherapy because of the natural course of the growing disease, and only randomized trials can answer the question of whether the phenomenon of hyperprogression is specific to immunotherapy treatments.

Some of these studies had reported clinical and biological predictive factors of hyperprogression, but their results are contradictory. Reported predictive factors were an age over 65 years [14], the presence of more than two metastatic sites [18], female gender [19], low PD-L1 expression by tumor cell and high density of TAM within the tumor [20], and regional recurrence for HNSCC patients [15]. Kato et al. also reported a shorter TTF under immunotherapy for patients with *EGFR* and *MDM2/4* alterations [21]. However, none of these observations were consistent across studies. In two studies, ECOG performance status of patients was not a predictive factor, but patients were almost all in good health with a performance status between 0 or 1 [18,19]. Hence, we cannot retain any of these variables as robust predictive factors, and further data are needed to identify which patients are more likely to experience hyperprogressive disease under immunotherapy.

Interestingly, there was no difference in hyperprogression rate reported between patients treated with anti-PD-1 and PD-L1 antibodies [14,15,18], but there are fewer data on other ICI and even more for other type of immunotherapy. Two studies only reported patients treated with monotherapy of PD-1 or PD-L1 inhibitors [14,18]. In the other studies, few patients were treated with a combination of PD-1/PD-L1 inhibitors and other immunotherapy (mostly CTLA-4 inhibitors) [19,22], except one patient treated with CTLA-4 inhibitor alone in Russo’s study [20] and in Kato’s study, where 22% of patients were treated with CTLA-4 inhibitor alone or in combination with PD-1/PD-L1 inhibitors and 11% with other immunotherapy but without reporting hyperprogression rate within this cohort [21]. Hence, we cannot exclude the hypothesis that hyperprogression is an immune checkpoint inhibitor-specific pattern of response among immune treatments.

Few studies, to our knowledge, assessed the question of hyperprogression potential biological mechanisms [20,34,39]. These preliminary translational studies enlighten how hyperprogression could occur among patients treated with immunotherapy and do support the hypothesis that hyperprogression could be an immune-related phenomenon. The potential mechanisms of hyperprogression remain to be characterized.

## 6. Practical Considerations

The frequency of hyperprogressive disease under immunotherapy should alert practitioners to the potential risks of these new treatments and the complexity of their use. Hyperprogressive patients had a significatively shorter OS compared to patients experiencing classical progression in two studies [18,22], suggesting that hyperprogression has a deleterious impact and that it should be managed as a therapeutic emergency. Several retrospective studies and case reports reported unexpected response rates to chemotherapy following progression under immunotherapy [43,44]. Thus, chemotherapy could represent a licit salvage treatment in cases of progression under immunotherapy. However, there is no specific data on chemotherapy response rate after hyperprogression, and it remains unknown whether chemotherapy could counterbalance the survival impact of hyperprogression. Data reported by Weiss et al. suggested that the measurement of chromosomal instability on plasma/serum-derived cfDNA between each immunotherapy cycle could help detect earlier hyperprogression [23]; however, this strategy needs to be further evaluated prospectively.

ICI are now commonly used in oncology, and implicate new ways to manage patient toxicities and response. New radiological criteria have been developed to include the concepts of unconfirmed progressive disease and pseudoprogression (Table 2), but RECIST evaluation remains the main criteria used in current practice and in clinical trials. Radiological evaluation alone is not enough to distinguish pseudoprogression from hyperprogression, since even the apparition of new lesions can be part of pseudoprogression with the immunotherapy-specific radiological criteria [11,12,13]. Furthermore, as described by Ferrara et al., a pure radiological definition of hyperprogression could lead to a misclassification of pseudoprogression as hyperprogression [18]. Thus, a clinical evaluation is essential to interpret radiological responses under immunotherapy. While immunotherapy could be maintained in cases of patients with radiological progression but clinical benefit in the hypothesis of pseudoprogression, patients with clinical aggravation, such as pain worsening or ECOG performance status decrease, or rapid progression should benefit, by contrast, from an earlier radiological evaluation (Figure 4). Patients experiencing clinical presentation suggesting hyperprogressive disease should not receive further ICI perfusions, should be reassessed early, and should be switched to another potential efficient treatment, such as salvage chemotherapy. Treatment should be interrupted in cases of hyperprogression, and patients should be reassessed to balance the likelihood of potential pseudoprogression, which is rare. We strongly recommend in cases of rapid progression to interrupt immunotherapy and rapidly re-assess the patient, which might allow switching to another treatment in patients with a still good clinical condition.

## 7. Conclusions

Hyperprogression is a frequent pattern of response with immunotherapy, reported in 4 to 29% of patients across retrospective studies, without strong predictive factors identified. While immunotherapies are increasingly used in oncology, there is a need to identify a consensual definition of hyperprogressive disease that would also include clinical criteria in order to be used by oncologists in current practice and in studies to homogenize results. More clinical, biological, and histopathological data are also needed to better understand the mechanisms of hyperprogression, identify clear predictive factors, and possibly prevent it.

## Figures and Tables

**Figure 1 ijms-20-02674-f001:**
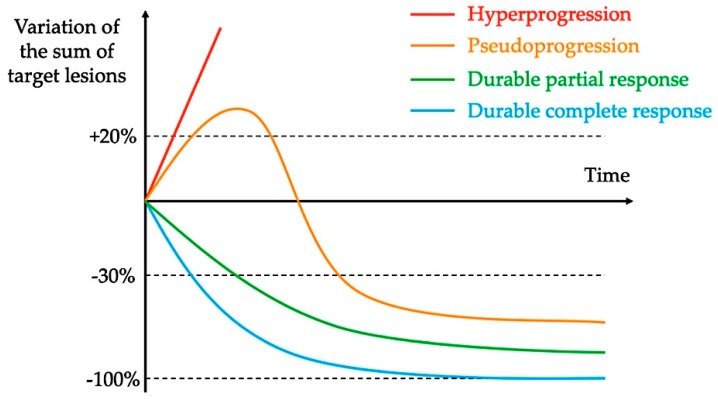
Examples of new patterns of response and progression with immunotherapy.

**Figure 2 ijms-20-02674-f002:**
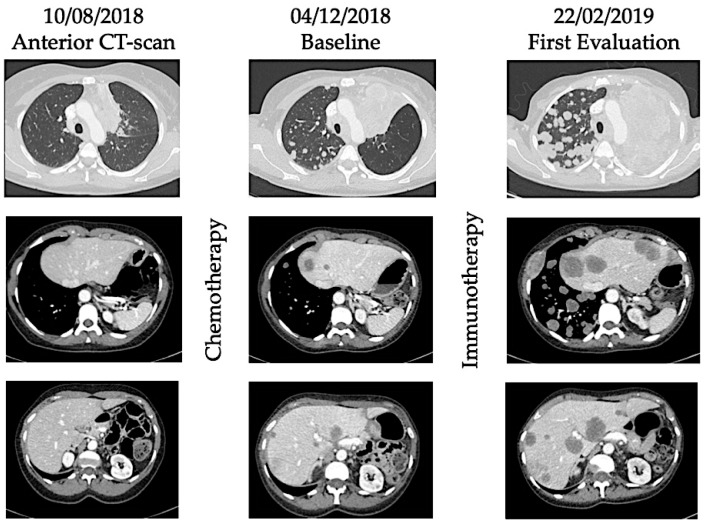
Illustration of hyperprogression under immunotherapy. A 53-year-old female patient with metastatic (lung) submandibular gland epidermoid carcinoma was treated in third line with weekly Methotrexate. After 4 months, patient experienced disease progression with appearance of lung, hepatic, and bone metastases. As fourth line, she received Nivolumab, an anti-PD-1 inhibitor. After 4 injections, she presented with major dyspnea with massive disease progression on computed-tomography (CT) scan. She died 64 days after immunotherapy initiation.

**Figure 3 ijms-20-02674-f003:**
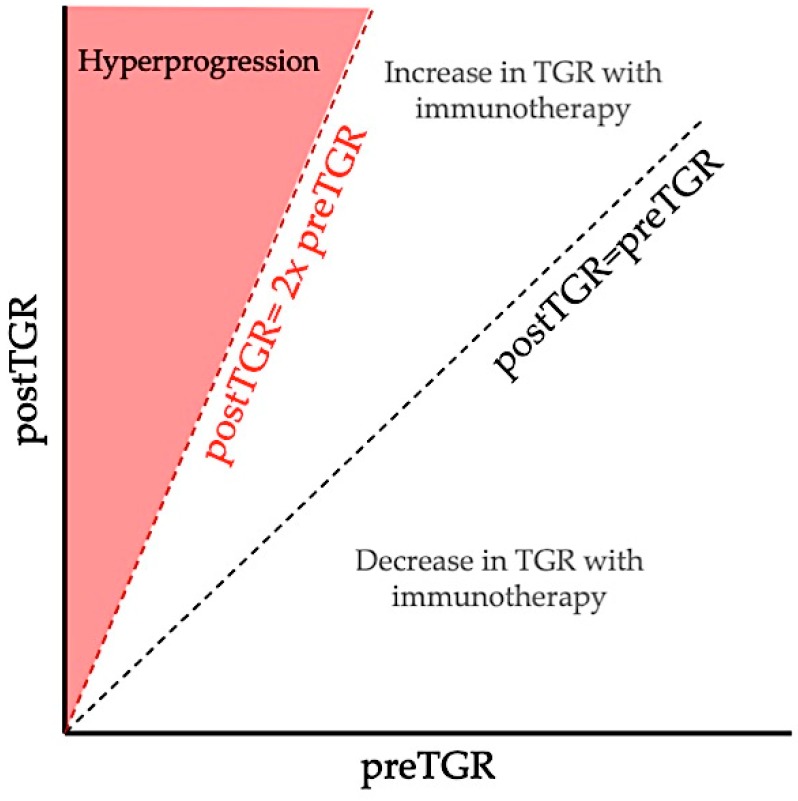
Definition of hyperprogression by Champiat et al. based on TGR variation. Comparison between pre-immunotherapy TGR (preTGR), computed with the anterior and baseline CT scan, and immunotherapy TGR (postTGR), computed with baseline CT scan and first evaluation during immunotherapy. Champiat defined hyperprogression as a RECIST progression at the first evaluation and at least two-fold TGR increase between pre-immunotherapy and immunotherapy period [14].

**Figure 4 ijms-20-02674-f004:**
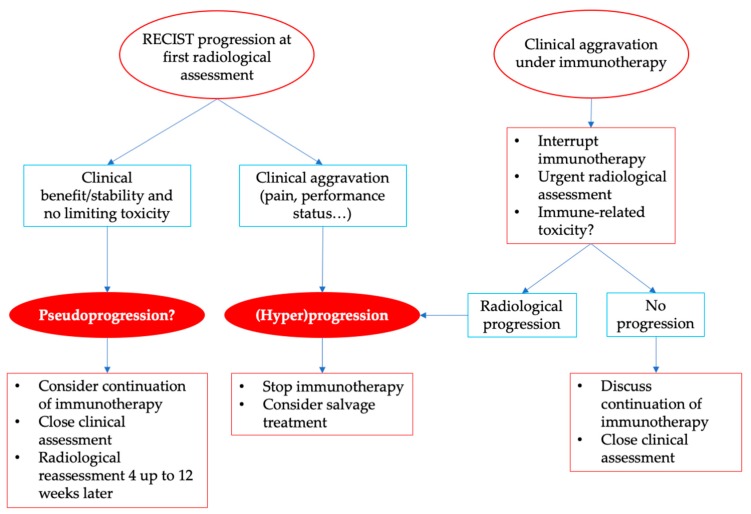
Management of radiological progression and clinical aggravation under immunotherapy.

**Table 1 ijms-20-02674-t001:** Rates of hyperprogression in patients receiving immune checkpoints inhibitors.

Study Drugs	Cancer Type	Definition of Hyperprogression	Number of Patients	Rates	Predictive Factors Identified	References
PD-1/PD-L1 inhibitors (phase 1 trials)	All cancer	RECIST progression **and** ≥2-fold increase TGR	131	9%	Age > 65 years old	[14]
PD-1/PD-L1 inhibitors	NSCLC	RECIST progression **and** ∆TGR > 50%	406	13.8%	>2 metastatic sites	[18]
ICI and/or costimulatory molecules (phase 1 trials)	All cancer	RECIST progression **and** ≥2-fold increase TGR	182	7%	Female gender	[19]
PD-1/PD-L1 inhibitors	HNSCC	≥2-fold increase TGK	34	29%	Regional recurrence in the irradiated field	[15]
ICI	NSCLC	RECIST progression **and at least 3 of:** TTF < 2 months **or** ≥50% increase of sum of target lesions major diameter **or** ≥2 new lesions in organ already involved **or** Spread to a new organ **or** ECOG PS ≥ 2	152	25.7%	Density of myeloperoxidase myeloid cells within the tumor Low PD-L1 expression in tumor cells	[20]
ICI or costimulatory molecules	All cancer	TTF < 2 months **and** >50% increase in tumor burden (irRECIST) **and** ≥2-fold increase in progression pace	155	4%	*EGFR*, *MDM2/4* and *DNMT3A* alterations	[21]
ICI (phase 1 trials)	All cancer	TTF < 2 months **and** ≥10mm increase in measurable lesions **and** ≥40% increase in target tumor burden or >20% plus appearance of multiple new lesions	214	15%		[22]

HNSCC, head and neck squamous cell carcinoma; ICI, immune checkpoint inhibitor; irRECIST, immune-related response-evaluation criteria in solid tumors; NSCLC, non-small-cell lung cancer; RECIST, response-evaluation criteria in solid tumors; TGK, tumor growth kinetics; TGR, tumor growth rate; TTF, time to treatment failure.

**Table 2 ijms-20-02674-t002:** Overview of immune-specific related response criteria reported in the literature.

	RECIST 1.1 [10]	irRC [11]	irRECIST [12]	iRECIST [13]
**Lesion measurement**	Unidimensional	Bidimensional	Unidimensional	Unidimensional
**Baseline lesion size**	≥10 mm	5 × 5 mm	≥10 mm	≥10 mm
**Baseline lesion number**	5 total, 2 per organ	10 total, 5 per organ	5 total, 2 per organ	5 total, 2 per organ
**CR**	Disappearance of all lesions	Disappearance of all lesions	Disappearance of all lesions	Disappearance of all lesions
**PR**	≥30% decrease from baseline	≥50% decrease from baseline	≥30% decrease from baseline	≥30% decrease from baseline
**SD**	Neither PR or PD	Neither PR or PD	Neither PR or PD	Neither PR or PD
**PD**	≥20% increase from nadir (≥5 mm)	≥25% increase from nadir	≥20% increase from nadir (≥5 mm)	≥20% increase from nadir (≥5 mm)
**Confirmed progressive disease**	Not applicable	At least 4 weeks after	At least 4 weeks after and up to 12 weeks	At least 4 weeks after and up to 8 weeks
**Appearance of new lesions**	Always PD	Incorporate in the sum of measurement	Incorporate in the sum of measurement	Unconfirmed progressive disease, not included in the sum of measurement

RECIST, response-evaluation criteria in solid tumors; irRC, immune-related response criteria; irRECIST, immune-related RECIST; iRECIST, immune RECIST; CR, complete response; PR, partial response; SD, stable disease; PD, progressive disease.

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
