# Peer review of "Hyperprogression under Immunotherapy"

_ijms, 2019, doi:10.3390/ijms20112674_

Reviewer 1 Report

The authors compose a review on hyperprogression under immunotherapy, defining the frequency, predictive factors, biological mechanisms and controversies of the hyperprogression phenomenon. The authors also elaborate on the inconsistent criteria used to define hyperprogression, as well as the discrepancy between different correlative results analyzed by various studies.  There is another recent review done on the topic by Champiat et al., referenced in the study, but this does add some additional references and depth to the topic. Despite the need for a comprehensive review on hyperprogression experienced when undergoing immunotherapy, there are a few minor concerns:

1.     When the authors describe hyperprogression (45), they make an association between unexpected rapid disease progression caused by immunotherapy and deleterious effects on cancer cell proliferation. Did the authors mean that it leads to cancer cell proliferation which is deleterious to the patient?

2.     A separate discussion between tumor pseudoprogression and hyperprogression is needed. The authors discuss how radiological imaging alone cannot be used to distinguish the two but do not give any more information regarding other methods used to confirm one or the other. Additionally, it would be enlightening to describe various radiological similarities between pseudoprogression and hyperprogression along with representative figures to aid in comprehension.

3.     The authors suggest a cessation or change in treatment following identification of hyperprogression, however they do not identify any studies that show hyperprogression can be reversed or halted following any intervention.

4.     Despite inconsistent findings regarding the Time to Treatment Failure (TTF) amongst different cancers, an explanation of the increased TTF in melanoma compared to other cancers would be insightful when associating different cancer characteristics with susceptibility to hyperprogression.

Author Response

POINT-BY-POINT REPLY

(our answers are in blue, and we have highlighted in yellow the changes introduced in the revised version of the manuscript)

 Reviewer 1:

 The authors compose a review on hyperprogression under immunotherapy, defining the frequency, predictive factors, biological mechanisms and controversies of the hyperprogression phenomenon. The authors also elaborate on the inconsistent criteria used to define hyperprogression, as well as the discrepancy between different correlative results analyzed by various studies.  There is another recent review done on the topic by Champiat et al., referenced in the study, but this does add some additional references and depth to the topic. Despite the need for a comprehensive review on hyperprogression experienced when undergoing immunotherapy, there are a few minor concerns:

 1.     When the authors describe hyperprogression (45), they make an association between unexpected rapid disease progression caused by immunotherapy and deleterious effects on cancer cell proliferation. Did the authors mean that it leads to cancer cell proliferation which is deleterious to the patient?

 We agree with the reviewer comment, and clarified our meaning (line 45) that immunotherapy could lead to cancer cell proliferation and thus to a deleterious effect for patients.

 Other studies also reported unexpected rapid disease progressions under immunotherapy, called hyperprogressions, suggesting that these treatments could have a deleterious effect and may lead to cancer cell proliferation or acceleration of progression pace.

 2.     A separate discussion between tumor pseudoprogression and hyperprogression is needed. The authors discuss how radiological imaging alone c

annot be used to distinguish the two but do not give any more information regarding other methods used to confirm one or the other. Additionally, it would be enlightening to describe various radiological similarities between pseudoprogression and hyperprogression along with representative figures to aid in comprehension.

 We thank the reviewer for this interesting comment. As pointed out, we added sentences in paragraph 6 to insist on the fact that radiological criteria alone are not sufficient to distinguish between hyperprogression and pseudoprogression.

To follow the reviewer’s suggestion, we also added a new figure (Figure 4) to illustrate how we propose managing radiological progression and clinical aggravation under immunotherapy.

 Radiological evaluation alone is not enough to distinguish pseudoprogression from hyperprogression, since even the apparition of new lesions can be part of pseudoprogression with the immunotherapy specific radiological criteria [11,12,13]. Furthermore, as described by Ferrara et al., a pure radiological definition of hyperprogression could lead to a misclassification of pseudoprogression into hyperprogression [18]. Thus, a clinical evaluation is essential to interpret radiological responses under immunotherapy.

 Figure 4. Management of radiological progression and clinical aggravation under immunotherapy.

3.     The authors suggest a cessation or change in treatment following identification of hyperprogression, however they do not identify any studies that show hyperprogression can be reversed or halted following any intervention.

 This is a really interesting comment, however this is difficult to address because for now there is indeed no published data that can guide the intervention to follow in case of observed hyperprogression.

The only suggestion we can make, in our opinion, in case of hyperprogression observed under immunotherapy, is to rapidly change the treatment if we observe a rapid symptomatic progression, as we would also do for any other type of treatment. In our opinion, it is important not to pursue immunotherapy in case of radiological progression if the patient deteriorates clinically, because it cannot be considered as a pseudoprogression in this case. For now, we only have data from retrospective studies reporting interesting response rates to chemotherapy following progression under immunotherapy (as explained lines 333 to 338), but we do not have the same data in case of hyperprogression under immunotherapy for now.

 4.     Despite inconsistent findings regarding the Time to Treatment Failure (TTF) amongst different cancers, an explanation of the increased TTF in melanoma compared to other cancers would be insightful when associating different cancer characteristics with susceptibility to hyperprogression.

 We thank the reviewer for this comment. We modified our sentences line 275-277, to precise that short TTF alone is not enough to define hyperprogression, and could be a marker of standard progression. For now there is no data suggesting that hyperprogression is less frequent among patients with melanoma.
However, Kato et al. reported a longer TTF in melanoma patients compared to other types of cancer, which may suggest that hyperprogression is less frequent among melanoma patients, even if a short TTF is not enough to define hyperprogression and could be a sign of other event like standard progression or limiting toxicity [21].

Reviewer 2 Report

Dear Authors,

I reviewed you article "hyperprogression under immunotherapy". An emerging phenomenon of importance and lacking attention in existing literature. Your article provides a good compilation of existing studies, even if the field will likely expand considerably and fast as new studies are available. Well written and organised, this review will be helpful to the community.

I suggest adding a sentence in one of sections 3, 4, 5 or 6 referring to the eventual added value of immunological characterisation (even PBMCs) of patients prior to immunotherapy in order to cross this information with outcome of treatment (including PBMC immunophenotype after ICI) , as there is a strong likelihood of immune related causality. 

One missing aspect is the reference to the existence or not of hyperprogression in the preclinical mouse models on the pathway to ICI therapy. A good mouse model of hyperprogression would be extremely useful in order to unveil the mechanisms underlying hyperprogression.

Minor aspect: please try in formatting to keep legend from table 2 in the same page as table. 

Best regards,

Author Response

 POINT-BY-POINT REPLY

(our answers are in blue, and we have highlighted in yellow the changes introduced in the revised version of the manuscript)

 Reviewer 2:

 Dear Authors,

I reviewed you article "hyperprogression under immunotherapy". An emerging phenomenon of importance and lacking attention in existing literature. Your article provides a good compilation of existing studies, even if the field will likely expand considerably and fast as new studies are available. Well written and organised, this review will be helpful to the community.

I suggest adding a sentence in one of sections 3, 4, 5 or 6 referring to the eventual added value of immunological characterisation (even PBMCs) of patients prior to immunotherapy in order to cross this information with outcome of treatment (including PBMC immunophenotype after ICI) , as there is a strong likelihood of immune related causality. 

 We thank the reviewer for this interesting comment and agree with this observation. As the reviewer points out, we added sentences in paragraph 4 to provide data regarding tumor immune infiltrate, immune characterization and response to immunotherapy.

 Several primary and adaptive mechanisms of resistance to ICI have been described. Response to immune checkpoint inhibitors seems to be conditioned by the infiltration of tumors by activated T cells, witnessing an ongoing active anti-tumor immune response. Tumors that lack TILs in the tumor bed, corresponding to so called “cold” tumors”, have been described as presenting low or no response to ICI [24–26]. The presence of immunosuppressive cells in tumor tissues like Tregs [27], or other myeloid-derived suppressor cells  [20] could also alter anti-tumor immunity.

Others mechanisms of resistance to ICI have been reported such as the absence of tumor recognition by T cells, due to the lack of immunogenic tumor antigens (i.e. low mutational burden, absence of neoantigens or cancer-testis antigens), or to the development by the tumor of mechanisms that impair antigen presentation like major histocompatibility complex down-regulation [28,29].

It has also been reported that alterations in tumor-intrinsic oncogenic pathways could impair the anti-tumor immune response, such as the activation of the PI3K/AKT/mTOR signalling pathway [30,31] or the WNT/β-catenin signaling pathway [32]. All these mechanisms of resistance should be further assessed in the context of hyperprogression.

 One missing aspect is the reference to the existence or not of hyperprogression in the preclinical mouse models on the pathway to ICI therapy. A good mouse model of hyperprogression would be extremely useful in order to unveil the mechanisms underlying hyperprogression.

 We agree with reviewer’s comment. We have cited in paragraph 4 two articles that investigated hyperprogression in mice models: the article of Russo et al, and the article of Stein et al.

We added in this paragraph another recently published reference that investigated the potential role of PD-1 blockade in facilitating the proliferation of highly suppressive PD-1+ effector T regulatory cells in hyperprogressive patients, resulting in the inhibition of antitumor immunity, and for which authors used a mouse model to validate their results.

 Recently, Kamada et al assessed pre and after anti-PD-1 treatment tumor samples from patients with advanced gastric cancer who experienced hyperprogression [39]. They found that PD-1 blockade increased tumor infiltrating proliferative effector Tregs, contrasting with their reduction in tumor samples of non-hyperprogressive patients. Under PD-1 blockade, they observed a significant enhancement of Tregs suppressive activity. In vivo, the genetic ablation of PD-1 or the antibody-mediated blockade of PD-1 in Tregs increased their proliferation and suppression of antitumor immune responses, suggesting that PD-1 blockade might facilitate the proliferation of highly suppressive PD-1+ effector Tregs in hyperprogressive patients, resulting in the inhibition of antitumor immunity.

 Minor aspect: please try in formatting to keep legend from table 2 in the same page as table. 

 As suggested by the reviewer, we have modified the format of the manuscript so that legend from table 2 appears in the same page as table 2.